# Integrated LCOS-SLM-Based Laser Slicing System for Aberration Correction in Silicon Carbide Substrate Manufacturing

**DOI:** 10.3390/mi16080930

**Published:** 2025-08-13

**Authors:** Heng Wang, Qiang Cao, Yuting Hou, Lulu Yu, Tianhao Wu, Zhenzhong Wang, Du Wang

**Affiliations:** 1The Institute of Technological Sciences, Wuhan University, Wuhan 430072, China; wangheng2021@whu.edu.cn; 2Shandong Engineering Research Center of New Optoelectronic Information Technology and Devices, School of Mathematics and Physics, Qingdao University of Science & Technology, Qingdao 266061, China; 3R&D Department, Suzhou Ultra Wafer Co., Ltd., Suzhou 215124, China; yuting_hou@163.com (Y.H.); yululu89@hotmail.com (L.Y.); tianhao_wu1217@163.com (T.W.); wangzz@mat-jitri.cn (Z.W.)

**Keywords:** silicon carbide, laser slicing, aberration correction, spatial light modulator, laser power attenuation

## Abstract

Silicon carbide (SiC), a wide-bandgap semiconductor, is renowned for its exceptional performance in power electronics and extreme-temperature environments. However, precision low-loss laser slicing of SiC is impeded by energy divergence and crack delamination induced by refractive-index-mismatch interfacial aberrations. This study presents an integrated laser slicing system based on a liquid crystal on silicon spatial light modulator (LCOS-SLM) to address aberration-induced focal elongation and energy inhomogeneity. Through dynamic modulation of the laser wavefront via an inverse ray-tracing algorithm, the system corrects spherical aberrations from refractive index mismatch, thus achieving precise energy concentration at wanted depths. A laser power attenuation model based on interface reflection and the Lambert–Beer law is established to calculate the required laser power at varying processing depths. Experimental results demonstrate that aberration correction reduces focal depth to approximately one-third (from 45 μm to 15 μm) and enhances energy concentration, eliminating multi-layer damage and increasing crack propagation length. Post-correction critical power measurements across depths are consistent with model predictions, with maximum error decreasing from >50% to 8.4%. Verification on a 6-inch N-type SiC ingot shows 90 μm damage thickness, confirming system feasibility for SiC laser slicing. The integrated aberration-correction approach provides a novel solution for high-precision SiC substrate processing.

## 1. Introduction

Silicon carbide (SiC), as a wide bandgap semiconductor material, exhibits great application potential in the fields of power electronics, radio frequency devices, and new energy vehicles due to its excellent high-temperature stability, high breakdown field strength, and high thermal conductivity [1,2,3]. At present, the popular AR glasses also use SiC as the lens material to achieve a performance leap [4], which further increases the demand for SiC. Although the growth technology of SiC crystals has developed rapidly in recent years, high-quality crystals still come at a high cost due to the strict requirements of the growth process [5,6,7]. Therefore, there is great value in improving the material utilization rate of SiC crystals. The process of cutting SiC ingots into substrates is currently at the stage where material loss is the most severe. Replacing traditional wire sawing with laser slicing is considered an effective method to reduce material loss and improve efficiency during the slicing process [8,9].

Therefore, the SiC laser slicing technology has received extensive attention, and a large number of studies have been conducted in various aspects such as the process, technology, and mechanism [10,11]. In 2017, Kim et al. used femtosecond laser double pulses, combined with the beam shaping technology, to cut thin slices from N-type silicon carbide crystals. The damage thickness was less than 24 μm, but the efficiency was extremely low, with a processing speed of 0.1 mm/s [12]. In 2023, Zhang et al. achieved the slicing of semi-insulating SiC through picosecond lasers, but the damage thickness reached more than 150 μm [13]. In 2025, Yao et al. systematically studied the influence of the processing direction and incident surface on the separation strength during the picosecond laser slicing process [14]. Through parameter optimization, the separation strength was reduced to 1.79 MPa, and the slicing of a 6-inch N-type SiC ingot was achieved. However, the damage thickness reached 120 μm. Currently, the main factor restricting the application of SiC laser slicing technology is the excessive damage thickness.

When lasers are used for internal processing of transparent materials, the interfacial aberration caused by the refractive index mismatch is one of the reasons for the large loss [15]. The wavefront shaping technology realized through a spatial light modulator can effectively eliminate aberrations and improve the precision of laser processing inside materials [15,16,17]. In common laser processing systems, a solution combining a static spatial light modulator (SLM) with the three-dimensional movement of the sample stage is adopted [14,16,18]. However, this design introduces a critical limitation: to ensure a fixed relative position between the spatial light modulator (SLM) and the objective lens, the Z-axis displacement system must be rigidly integrated into the motion platform. This integration inevitably increases the total load mass, which not only restricts the acceleration response ability of the motion stage but also reduces the dynamic stability during the motion process, making it difficult to meet the requirements for both efficiency and stability in slicing processing.

Furthermore, due to the extreme sensitivity of SiC crystal growth to process conditions, even minor process fluctuations can induce doping concentration variations between ingots [7], leading to significant variations in laser transmittance. To ensure consistency in laser slicing processes, dynamic adjustment of laser processing parameters based on individual ingot characteristics is required, which remains a key challenge in current SiC laser slicing technology.

To address the aforementioned challenges, this study proposes an integrated laser slicing system utilizing a liquid crystal on silicon spatial light modulator (LCOS-SLM). By combining wavefront shaping via the LCOS-SLM with an inverse ray tracing algorithm, refractive index mismatch-induced aberrations are corrected, enabling precise energy concentration at wanted depths. Furthermore, a power compensation model is developed to quantify transmittance variations and depth-dependent attenuation across individual SiC ingots, providing critical inputs for dynamic laser parameter optimization to ensure uniform processing outcomes. This paper comprehensively details the system’s optical design, aberration correction principles, and experimental validation, offering an innovative approach to high-efficiency, high-quality SiC substrate fabrication.

## 2. System Design

To address the challenges of refractive index mismatch-induced aberration and depth-dependent energy attenuation, the proposed system integrates an LCOS-SLM for dynamic wavefront shaping and a power compensation model to ensure consistent energy delivery at wanted depths. The following sections detail the optical and mechanical design of the system. The system employs a reflective liquid crystal on silicon phase-only spatial light modulator (LCOS-SLM, X15223-03R, Hamamatsu Photonics K.K., Hamamatsu, Japan). Its active area measures 15.9 mm × 12.8 mm, corresponding to a 1272 × 1024 pixel array with a pixel pitch of 12.5 μm and a fill factor of 96.8%. Each pixel can be independently addressed through 8-bit digital signals transmitted from a computer via a Digital Visual Interface (DVI). This LCOS-SLM achieves pure phase modulation from 0 to 2π in the near-infrared band (1000–1100 nm), with 256 gray levels (0–255) linearly mapped to phase modulation values. Phase wrapping technology is implemented to extend phase control beyond single-cycle limitations, enabling broader-range phase manipulation. The system utilizes a 1064 nm nanosecond laser source with a pulse width of 1 ns.

As shown in Figure 1a, the reflective SLM requires the incident beam to form an angle of less than 10° with the surface normal. Additionally, to ensure optimal modulation performance and resolution, the laser beam is expanded to a diameter of 9 mm (1/e^2^ intensity profile) to maximize pixel coverage. The polarization state of the incident light is aligned with the long-axis orientation of the liquid crystal molecules via a half-wave plate, achieving maximum phase modulation efficiency. The objective lens has a numerical aperture (NA) of 0.65, a focal length of 4 mm, and an entrance aperture of 5.2 mm (calculated as 0.65 × 4 × 2). A 4f system with lenses of 75 mm and 30 mm focal lengths (demagnification ratio: 2.5) is incorporated to project and downsize the SLM-modulated beam into the objective lens, resulting in an effective pixel size of 5 μm after projection.

The system adopts an integrated design, incorporating mirrors, the SLM, the 4f system, and the objective lens into a unified assembly. The 3D CAD model and physical photograph of the system are depicted in Figure 1b,c. The main structure, fabricated from 6061 aluminum alloy with dimensions of 340 mm (height) × 170 mm (width), ensures mechanical stability and optical alignment through precision machining. Mirror M1 directs the modulated beam vertically to maintain process consistency across varying depths, while mirror M2 directs the beam to the SLM at a controlled 10° incidence angle. The SLM-reflected beam remains strictly vertical and coaxial with both the 4f system and the objective lens to prevent beam distortion. All mirrors use Corning 7980 fused silica substrates with a reflectivity exceeding 99.8%. A laser displacement sensor (CL-P007, KEYENCE, Osaka, Japan, accuracy <1 μm) is integrated into the system. Its pre-calibrated positional relationship with the objective lens provides real-time surface height data for the processed sample.

The integrated design endows the system with exceptional modular portability. For laser slicing applications, the system employs an independent Z-axis (vertical motion) to accommodate samples of varying heights, achieving machining trajectory control through coordinated operation with external XY motion platforms. With the SLM and objective lens consolidated into an integrated optical module, precise focusing depth positioning is achieved solely through Z-axis adjustment. This plug-and-play configuration enables rapid deployment of the entire optical system as a self-contained unit onto motion platforms. By decoupling the dynamic loads between the Z-axis and XY axes, it effectively reduces inertial disturbances on the XY translation stage, thereby significantly enhancing dynamic stability during multi-axis processing.

## 3. Results and Discussion

### 3.1. Aberration Correction Principle

When a parallel light beam is focused by an objective lens in air, ideally, the light spot is elliptical and converges at the geometric focal point. Figure 2a shows the simulation results of the focusing of light with a wavelength of 1064 nm through an objective lens with a numerical aperture (NA) of 0.65 and a focal length of 4 mm in air (assuming the refractive index is 1). The light spot is elliptical, and the depth of focus is approximately 5 μm. However, when the light beam enters from air into a medium for focusing, the refraction effect causes an optical path difference for the rays with different angles of incidence, significantly changing the focusing characteristics [Figure 2b]. The focusing position will shift downward, and the amount of this downward shift can be approximately estimated by the product of the refractive index and the focusing depth. Figure 2c is the simulation result of the light focusing inside a SiC crystal (with a refractive index of 2.6), where the nominal focal point is set at a depth of 100 μm from the upper surface. The focused light spot shows a comet-tail-like distribution. The energy is mainly concentrated in the top region 180 μm below the nominal focal point. In addition, there are local energy concentration points in the tail, and the longitudinal length is greater than 50 μm. This phenomenon of energy dispersion will lead to a decrease in the longitudinal resolution of laser processing and an expansion of the damaged area, which is extremely detrimental to the laser slicing process.

The inverse ray tracing algorithm is used to calculate the aberration correction hologram. This method has significant advantages in correcting aberrations at large depths [19], and its principle is shown in Figure 2b. The arc in the diagram represents a spherical reference surface with a curvature radius *f*, which passes through the rear principal point F of an ideal objective lens with focal length *f*. The rear principal point F is located at the intersection of the spherical reference surface and the optical axis. Point I denotes the intersection point of the optical axis with the interface between the two media. When a plane wave is incident on the ideal objective lens, the output wavefront coincides with the reference surface and converges at the nominal focal point O. The distance from O to the medium’s upper surface is defined as the nominal depth *d*. The wanted focal point O’ lies at a depth *d_w_* below the medium’s surface. In the diagram, *θ* represents the incident angle of a light ray, *θ*_1_ denotes the incident angle required to focus the beam at point O’, and *θ*_2_ is the corrected refractive angle of the same ray. *n*_1_ and *n*_2_ correspond to the refractive indices of the materials on either side of the interface. The optical path from any point on the reference plane to the wanted focal point *O*’ is expressed as follows:(1)$$s\left(\theta \right)=n_{1}\times \bar{AB}+n_{2}\times \bar{BO^{\prime }}=n_{1}\times \frac{\left(Z_{A}-d\right)}{\cos\theta_{1}}+n_{2}\times \frac{d_{w}}{\cos\theta_{2}}$$
where *Z_A_* is the height difference between point A and point O. Thus, $s\left(\theta \right)-s\left(0\right)$ is defined as the optical path difference requiring compensation for rays with an incident angle *θ*. For a given optical system, the value range of *θ* is jointly determined by the numerical aperture of the objective lens and the focal length *f*. The relationship between *θ*_1_ and *θ*_2_ satisfies Snell’s law, and a unique solution exists for any *θ*. The phase that needs to be corrected can be calculated using the optical path difference:(2)$$\Psi =-\frac{2\pi }{\lambda }\left[s\left(\theta \right)-s\left(0\right)\right]$$
where λ is the wavelength of light. In practical applications, due to the limited phase modulation capability of the SLM, the phase difference must undergo phase wrapping—that is, being projected onto the 0–2π phase range. The final phase loaded onto the SLM is as follows:(3)$$\Psi_{\mathrm{SLM}}=\mathrm{mod}(\Psi ,2\pi )$$

However, due to the limited resolution of the SLM, an excessively large phase modulation range before phase wrapping will increase phase jumps after wrapping. This compromises the smoothness of the spatial phase distribution, leading to discontinuous phase transitions in the wavefront of the shaped optical wave and reducing diffraction efficiency. Therefore, we aim to minimize the phase modulation range, which can be quantified using the PV value—defined as the peak-to-valley difference between the maximum and minimum values of the modulation phase [15,19].

Under specific processing conditions, once the nominal focal depth (*d*) of the objective lens is fixed, aberration correction can theoretically be achieved at any arbitrarily set wanted depth (*d_w_*). However, variations in the wanted depth significantly impact the PV value. As shown in Figure 3, when the nominal depth is 100 μm, the holograms and their PV values are presented for the wanted depths of 100 μm, 200 μm, 300 μm, and 400 μm. It is evident that as the wanted depth increases from 100 μm to 400 μm, the PV value first decreases and then increases. With the rise in PV value, phase discontinuities in the hologram become increasingly pronounced. Areas with larger periodic variations generate stronger zero-order diffraction, which obscures the target image and reduces the signal-to-noise ratio. Therefore, it is essential to establish an optimal relationship between the wanted depth and nominal depth to achieve optimal aberration correction.

For a given *NA* and medium refractive index *n*_2_, the optimal defocus factor *s* (defined as the ratio of *d*/*d_w_*) can be determined by formula [15]:(4)$$s=\frac{n_{2}-\sqrt{n_{2}^{2}-{NA}^{2}}}{n_{1}-\sqrt{n_{1}^{2}-{NA}^{2}}}$$
for *n*_2_ = 2.6 and *NA* = 0.65, *s* ≈ 0.3442. Figure 4a displays the theoretical aberration-corrected hologram (1024 × 1024 pixels) designed for SiC crystal processing, with parameters set to a wanted depth *d_w_* = 200 μm and nominal depth *d* = 69 μm. Figure 4b simulates the correction effect: after aberration correction, the depth of focus is reduced to approximately one-third of its original value (cf. Figure 2c), and the longitudinal energy distribution becomes significantly more concentrated. Since hologram computation depends solely on objective lens parameters and is independent of laser spot size, spatial matching between the objective’s entrance aperture (diameter: 5.2 mm) and the SLM’s active area must be ensured during hologram loading. After applying a 2.5× demagnification via the 4f system, the projected diameter of the objective aperture onto the SLM plane expands to 13 mm. To address this overshoot relative to the SLM’s effective width (12.8 mm), the hologram is rescaled by a factor of 13/12.8. Furthermore, when the laser illuminates the SLM at a 10° incidence angle, elliptical distortion of the projected spot must be compensated. Based on geometric optics, the major axis of the elliptical spot scales by a factor of 1/cos 10° relative to the original diameter. The hologram adjusted with this correction is shown in Figure 4c.

### 3.2. Experimental Results

As shown in Figure 5a, the experimental samples were sourced from a 360 μm-thick N-type 6-inch SiC substrate (TanKeBlue, Beijing, China), diced into rectangular chips via diamond wire sawing. The laser processing configuration, as illustrated in Figure 5b, employs synchronized scanning and feed motions to achieve full-surface machining coverage. The sample surface height is determined using the laser displacement sensor, and the objective’s focal point is positioned at the surface via a pre-calibrated relationship. The objective is then translated along the z-axis to shift the laser focus into the sample’s interior. To reduce the interference of crystal anisotropy, the scanning direction aligns with the <−1100> crystal direction to ensure that the processing is performed on the same (0001) crystal plane. The feed direction follows <11−20>, with laser incidence normal to the C-face. Processing parameters include a laser power of 0.3 W, repetition rate of 50 kHz, scan spacing of 200 μm, scan speed of 100 mm/s, and nominal depth of 60 μm.

The processed sample exhibits distinct bright white stripes when observed at an inclined angle [Figure 5a], indicative of internal crack formation, enhancing localized light reflection. The processed sample was observed using an optical microscope (OMT-900HC, OUMIT, Suzhou, China), with the top view shown in Figure 5c. The damage zone formed by laser scanning exhibits a width of approximately 10 μm, flanked by cracks on both sides of the scanning track, each approximately 20 μm in length.

The sample was subsequently fractured perpendicular to the scanning direction (indicated by the blue arrow in Figure 5b) for cross-sectional analysis. The fractured samples were initially etched with potassium hydroxide (KOH) to delineate damage zones [20], followed by ultrasonic cleaning and subsequent fracture surface characterization using scanning electron microscopy (SEM, Phenom ProX, Phenom-World BV, Eindhoven, Netherlands). Figure 5d,e presents the machining outcomes without and with aberration correction, respectively. Consistent with optical microscopy observations, laser processing induced material damage and crack formation. Under laser irradiation, SiC undergoes amorphization and decomposition, inducing localized volumetric expansion. Given that the (0001) lattice plane of 4H-SiC exhibits the lowest cleavage energy [21], the induced expansion stress preferentially releases along this plane, generating cleavage cracks. Furthermore, due to the 4° off-axis angle between the N-type 4H-SiC’s lattice plane and its geometric surface [22], the resultant cracks maintain a 4° angular offset relative to the surface. Under uncorrected conditions (Figure 5d), the damage zone comprises two distinct crack layers spanning approximately 45 μm. With aberration correction (Figure 5e), however, a single crack forms, with its longitudinal damage span significantly reduced to approximately 15 μm (about one-third of the uncorrected state), while exhibiting greater lateral extension.

The experimental observations align closely with the simulation outcomes. Specifically, the energy distribution at the tip and tail of the comet-shaped spot in Figure 2c shows high consistency with the experimentally observed two-layer damage distribution. After aberration correction, the energy concentration significantly improves, as demonstrated by the simulated Gaussian-shaped spot in Figure 4b. The spatial distribution of laser intensity satisfies the following formula:(5)$$I\left(r,\;z\right)=I_{0}\cdot e^{-\frac{2r^{2}}{w{\left(z\right)}^{2}}}$$
where *w(z)* represents the beam radius as a function of the longitudinal position *z*. For silicon carbide (SiC), a damage threshold exists: when the local laser intensity exceeds the critical value *I_c_*, the material undergoes melting, amorphization, and decomposition [23]. The volume of the decomposition zone can be calculated by integrating over the region where the laser intensity exceeds *I_c_*:(6)$$V=\int_{0}^{z_{max}}\int_{0}^{r_{d}\left(z\right)}2\pi rdrdz$$

Here, *r_d_(z)* and *z_max_* are defined as the transverse and longitudinal boundaries where the laser intensity exceeds the critical value *I_c_*. After aberration correction, the energy distribution becomes more concentrated, resulting in expanded boundary ranges of *r_d_(z)* and *z_max_*, and consequently increasing the volume of the decomposition zone. Due to the constrained non-uniform expansion within the material, localized residual stress is significantly elevated, with enhanced stress concentration at crack tips, ultimately leading to increased crack propagation lengths. These results validate the feasibility of the proposed aberration correction method in suppressing aberrations and improving machining precision for SiC laser slicing.

### 3.3. Depth-Dependent Power Compensation Model

As laser energy is absorbed during propagation through SiC crystals, beam intensity attenuates with depth, necessitating distinct processing powers at different depths. This section examines the laser power attenuation pattern post-aberration-correction and develops a depth-dependent power compensation model. When the laser is incident from air onto the sample surface, reflection, scattering, and transmission occur. For smooth surfaces, scattering is generally negligible. Under normal incidence, the power reflectance *R* is given by the following:(7)$$R={\left(\frac{n_{2}-n_{1}}{n_{2}+n_{1}}\right)}^{2}$$
where *n*_1_ and *n*_2_ represent the refractive indices of the two media, respectively. Taking the refractive index of air as *n*_1_ = 1 and that of SiC as *n*_2_ = 2.6, the reflectance under normal incidence is calculated to be 19.75%. After entering the sample, the laser intensity attenuates with increasing penetration depth due to material absorption, a phenomenon that can be described by the Lambert–Beer law:(8)$$I=I_{0}e^{-\alpha z}$$
where *α* denotes the absorption coefficient, *I*_0_ the initial intensity, and *z* the penetration depth. Because the laser power is proportional to the light intensity. Given the known substrate thickness *h*, the absorption coefficient can be calculated by measuring the incident power *P_in_* before the substrate and the transmitted power *P_out_* after the substrate using a power meter, as shown in Figure 6a. When the laser is normally incident from air onto the substrate, considering a single reflection at both the upper and lower interfaces, the following equation is derived:(9)$$P_{out}=P_{in}\times \left(1-R\right)\times e^{-\alpha h}\times \left(1-R\right)=P_{in}{\left(1-R\right)}^{2}e^{-\alpha h}$$

By defining the transmittance *T* as the ratio of the transmitted light intensity (*P_out_*) to the incident light intensity (*P_in_*), the absorption coefficient *α* can be calculated using the following formula:(10)α=lnPout(Pin×1−R2)h=−1hlnT1−R2

Experimental measurements show that the transmittance of the sample for the 1064 nm laser wavelength is 21.39% (360 μm thickness), with an absorption coefficient of 30.5 cm^−1^.

Direct measurement of laser intensity at varying depths within the crystal is challenging. We therefore indirectly assess the intensity magnitude by observing the critical power for crack coalescence. As shown in Figure 6b, adjacent track cracks remain independent at lower laser powers. When the power increases to a threshold value, cracks from neighboring tracks coalesce (Figure 6c). Crack propagation requires the energy release rate at the crack tip to exceed a critical value [24]. Assuming a constant critical energy release rate for identical samples, the critical power for crack coalescence serves as an indirect indicator of laser–SiC interaction equivalence. The experiment maintained a laser frequency of 50 kHz, spacing of 100 μm, and scanning speed of 100 mm/s, while progressively increasing single-pulse energy until crack coalescence occurred. The critical power measurement resolution reached 0.12 μJ (6 mW). Processing depths were verified through cross-sectional observations. In Figure 7, red and yellow curves represent critical power values with and without aberration correction at different depths, respectively. At identical depths, aberration-corrected conditions require lower critical power compared to non-corrected scenarios, with this difference becoming more pronounced at greater depths.

Currently, unavoidable variations in transmittance exist among individual N-type SiC ingots due to doping concentration differences during crystal growth. To circumvent direct transmittance measurements of the ingots, a power prediction method based on low-depth data is proposed. Accounting for upper surface reflection and the proportional relationship between laser power and intensity, the power at depth *d* can be formulated by integrating the Lambert–Beer law:(11)$$P\left(d\right)=P_{0}\cdot \left(1-R\right)\cdot e^{-\alpha d}$$

At a processing depth of 100 μm, the critical power (incident power) is 240 mW, corresponding to an effective power of 141.9 mW at the 100 μm depth. Given the target power *P_w_* = 141.9 mW, the incident power *P*_0_ can be back-calculated using the following relationship:(12)$$P_{0}\left(d\right)=\frac{P_{w}}{\left(1-R\right)\cdot e^{-\alpha d}}$$

The calculated incident power required at different depths, based on the above equation, is plotted as the blue curve in Figure 7 Experimental results closely match the model predictions, showing a maximum error of 8.4% in critical power measurements under different processing depths. Discrepancies may arise from factors such as laser scattering, interface roughness, and transmittance variations due to doping concentration fluctuations within SiC ingots.

In the absence of aberration correction, the measured critical power deviates significantly from the model prediction (>50%). This occurs because aberration effects become increasingly pronounced with greater processing depths (Figure 2b,c), causing significant laser energy divergence. As shown in Figure 8a, at a nominal depth of 86 μm without aberration correction, the longitudinal span of laser-induced damage traces reaches 50 μm. In contrast, aberration-corrected processing (Figure 8b) maintains tightly focused damage confined to a single point, with damage spans remaining below 20 μm. The enhanced energy concentration under aberration correction directly reduces the required critical power compared to non-corrected conditions. These results not only validate the accuracy of aberration correction but also deliver a computational model for depth-dependent laser power requirements.

### 3.4. Ingot Processing Validation

To further verify the effectiveness of the laser slicing system, processing validation was conducted on a 6-inch N-type SiC ingot. The processing flow is illustrated in Figure 9. First, the ingot surface was ground, and the thickness after grinding was recorded. Laser processing was then performed with a laser power of 0.7 W, repetition frequency of 50 kHz, scan spacing of 150 μm, and scan speed of 100 mm/s. Separation of the substrate from the ingot was achieved using a universal testing machine. Both the separated substrate and the remaining ingot underwent subsequent grinding. The resulting substrate thickness was 230 μm, while the ingot thickness was reduced by 320 μm from the original ingot, indicating a laser processing damage thickness of 90 μm.

Compared to the observation results from small-area samples (Figure 5e: 15 μm, Figure 8b: <20 μm), the 6-inch ingot exhibited greater damage (90 μm). This is attributed to non-uniform transmittance caused by variations in ingot surface thickness and doping concentration. Table 1 presents a comparative analysis of the experimental results of SiC laser slicing reported in recent years. In comparison, our work has great advantages.

## 4. Conclusions

This study demonstrates the successful development and validation of an integrated LCOS-SLM-based laser slicing system specifically designed to overcome the critical challenges of spherical aberration and depth-dependent energy attenuation in silicon carbide (SiC) substrate manufacturing. The core innovation lies in the synergistic combination of dynamic wavefront correction utilizing a liquid crystal on silicon spatial light modulator (LCOS-SLM) and an inverse ray tracing algorithm, coupled with a depth-dependent laser power compensation model. This integrated approach was consolidated into a compact, modular optical assembly. Key achievements include the following:(1)The inverse ray tracing method enabled the calculation of optimized hologram patterns for the LCOS-SLM. Simulation results confirmed that this correction effectively suppresses refractive index mismatch-induced spherical aberration, restoring a tightly focused elliptical spot profile and reducing the focal depth to approximately one-third of its uncorrected value.(2)Experimental verification on N-type 4H-SiC samples demonstrated a significant reduction in longitudinal damage span, from approximately 45 μm without correction to about 15 μm with correction. This threefold improvement in energy concentration eliminates multi-layer damage while enhancing lateral crack propagation, a critical factor for efficient substrate separation.(3)A laser power attenuation model, incorporating interface reflection loss and material absorption, was established and experimentally validated. Post-aberration-correction critical power measurements across various depths showed excellent agreement with model predictions, reducing the maximum deviation from >50% (uncorrected) to 8.4%.(4)Validation on a full 6-inch N-type SiC ingot resulted in a damage thickness of 90 μm, confirming the practical feasibility of the integrated system for low-damage laser slicing of SiC substrates.

Future work prioritizes real-time laser parameter regulation coupled with ingot property pre-characterization to further minimize damage thickness. The system exhibits potential applicability to other wide-bandgap semiconductors (e.g., GaN, Ga_2_O_3_, diamond) and transparent materials requiring internal processing, such as optical waveguides and color centers, owing to its adaptive wavefront correction and power compensation capabilities.

## Figures and Tables

**Figure 1 micromachines-16-00930-f001:**
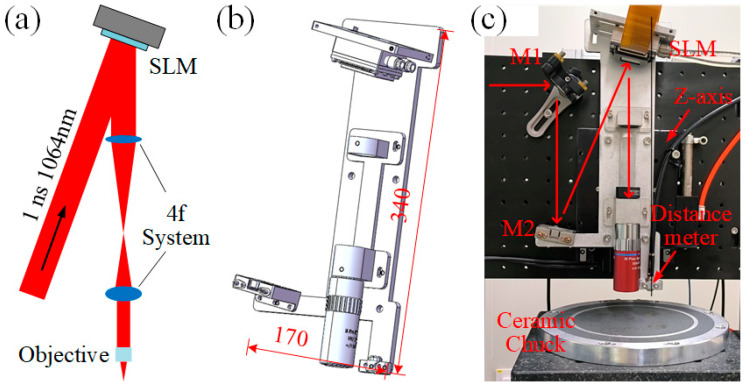
Integrated optical system. (**a**) Schematic diagram. (**b**) 3D CAD model. (**c**) Physical photograph.

**Figure 2 micromachines-16-00930-f002:**
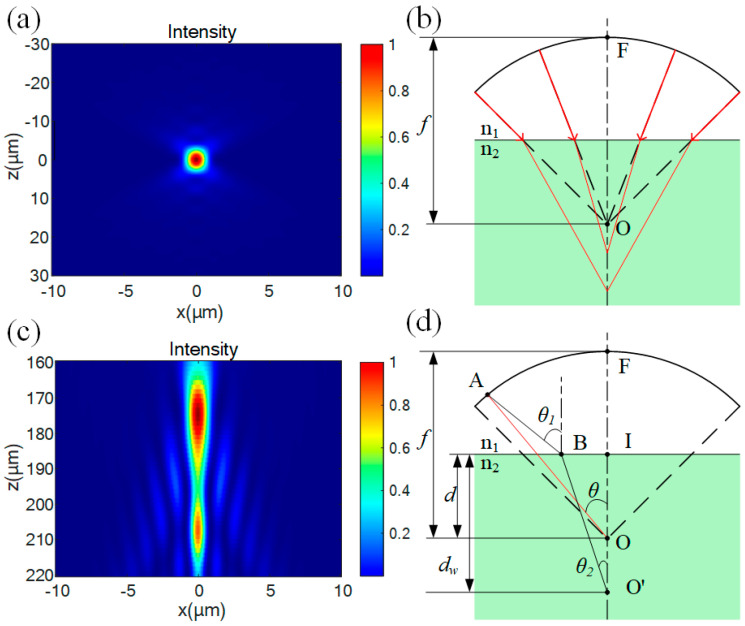
Focusing simulation results and schematic diagrams. (**a**) Simulation diagram of the focused light spot in air. (**b**) Schematic diagram of the aberration caused by the refractive index mismatch. (**c**) Simulation diagram of the focused light spot inside SiC. (**d**) Schematic diagram of the principle of aberration correction.

**Figure 3 micromachines-16-00930-f003:**
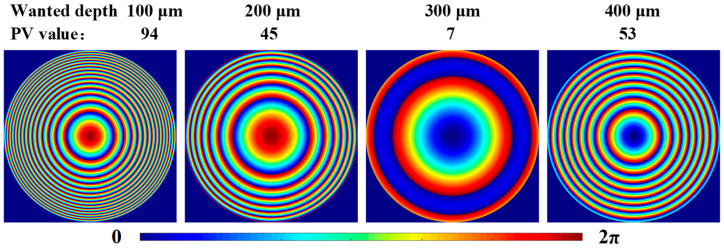
Holograms and PV values at various wanted depths with a nominal depth of 100 μm.

**Figure 4 micromachines-16-00930-f004:**
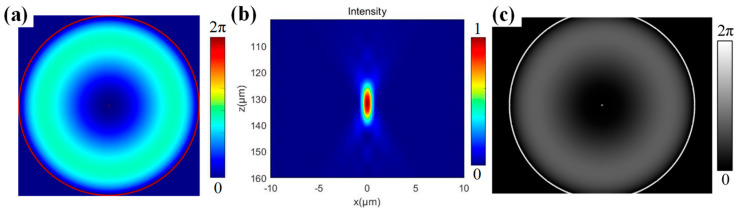
Holograms and simulation results. (**a**) Computed theoretical hologram. (**b**) Simulated result with aberration correction. (**c**) Experimentally loaded hologram.

**Figure 5 micromachines-16-00930-f005:**
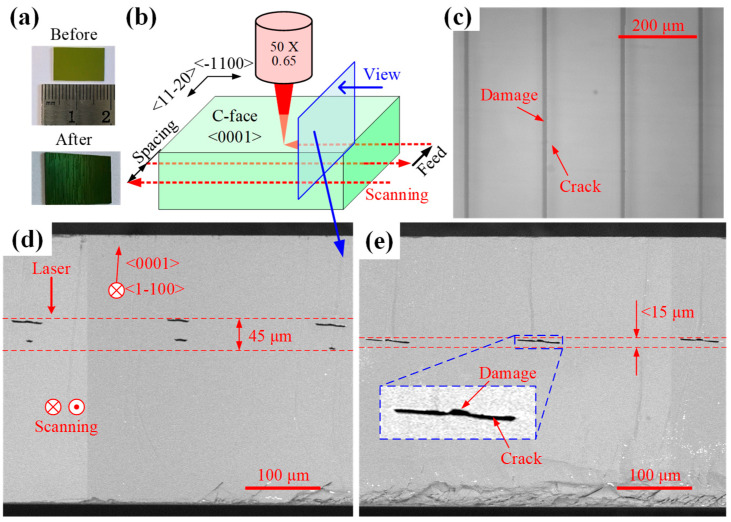
Experimental materials, methodology, and results. (**a**) Experimental sample. (**b**) Laser processing configuration. (**c**) Top-view optical micrograph. (**d**,**e**) Cross-sectional SEM images.

**Figure 6 micromachines-16-00930-f006:**
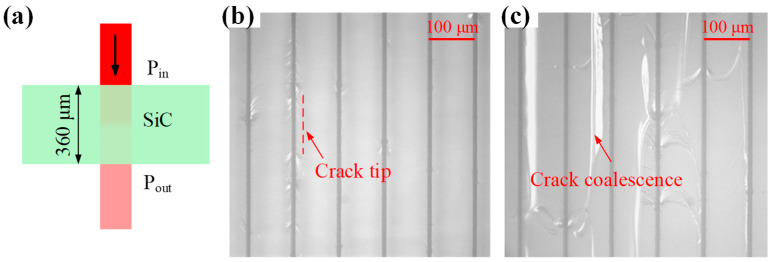
Transmittance measurement and critical power judgment. (**a**) Schematic diagram of transmittance measurement. (**b**) Optical microscopy image showing independent cracks and (**c**) merged crack morphology.

**Figure 7 micromachines-16-00930-f007:**
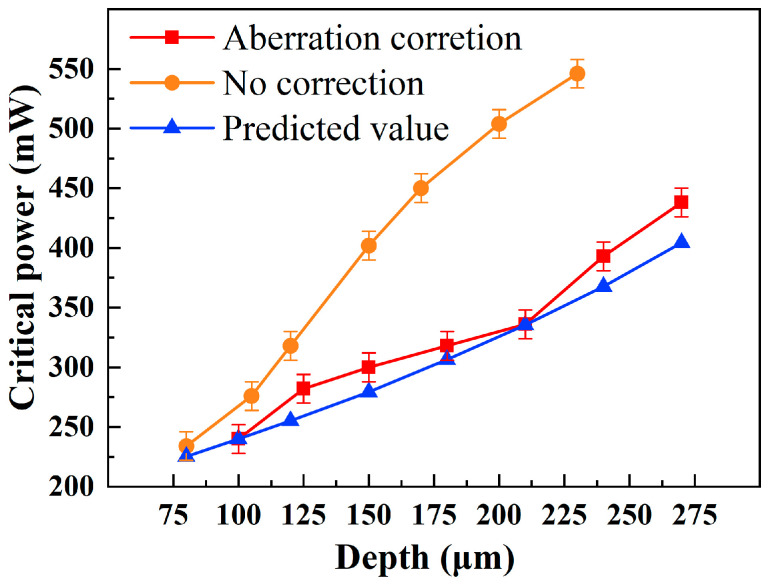
Experimental and predicted values of critical power at varying depths.

**Figure 8 micromachines-16-00930-f008:**
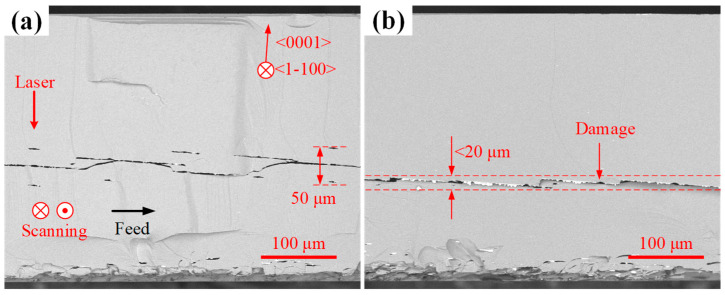
Cross-sectional views of samples processed at critical power under 86 μm nominal depth. (**a**) Without aberration correction. (**b**) With aberration correction.

**Figure 9 micromachines-16-00930-f009:**
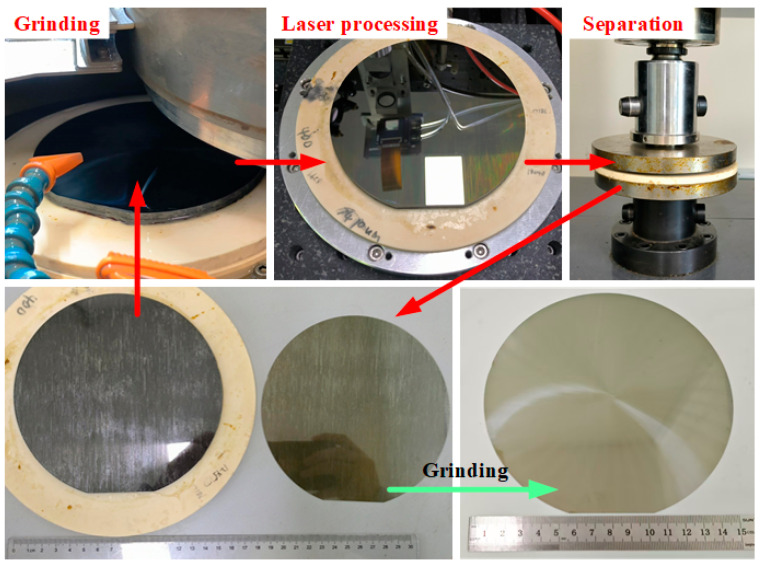
6-inch ingot processing validation.

**Table 1 micromachines-16-00930-t001:** Performance comparison of laser slicing for SiC wafers.

Year	Pulse Width (ps)	Damage Thickness (μm)	Ref.
2023	15	150	[13]
2025	15	120	[14]
2025	1000	90	This work

## Data Availability

The original contributions presented in the study are included in the article, further inquiries can be directed to the corresponding author.

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
