# Peer review of "Integrated LCOS-SLM-Based Laser Slicing System for Aberration Correction in Silicon Carbide Substrate Manufacturing"

_micromachines, 2025, doi:10.3390/mi16080930_

Round 1
Reviewer 1 Report
Comments and Suggestions for Authors
- Any ehancing of the data (microscope picture [Figure 5c; 6b,c;] is not acceptable "To enhance crack clarity...[line 246]", I think. Possible just show the actual place or borders on the picture by arrow or thin line... not to cover entire appearence by thick stripe.
- Not clear (not explaned) the origin of vertical crack (along laser beam direction) above each damage place in Figure 5e. Could you, please, clarify the situation? Is it the crack along scaning trajectory?
Reviewer 2 Report
Comments and Suggestions for Authors
This study presents an integrated LCOS-SLM-based laser slicing system for SiC substrate manufacturing, which corrects aberrations via dynamic wavefront modulation and a power compensation model, achieving reduced focal depth, enhanced energy concentration, and feasible application. Experimental results validate that the system significantly improves slicing precision with a maximum error reduction in critical power measurements and effectively addresses challenges in SiC laser slicing. The manuscript shows notable innovation and practical value, with relatively solid experimental data and credible conclusions. However, key details require supplementation and phrasing optimization before acceptance. I have the following questions/comments:
Comments 1: All image labels and scale bars in the paper should be consistent. Scale bars should be added to Figure 5(d) and Figure 8(a). Figure 4(c) lacks a phase scale.
Comments 2: How is the laser processing direction determined? Does the laser processing direction have an impact on damage?
Comments 3: The terms "wanted depth" and "target depth" are used interchangeably.
Comments 4: The notations in Figure 5(d) should maintain consistency with the main text, using uniform standards for upright and italic fonts.
Comments 5: Reference [25] in Table 1 is incorrect.
Comments 6: Reference [14] also involves an SLM system; what are the differences from this paper?
